# Changes in the secondary compounds of persimmon leaves as a defense against circular leaf spot caused by *Plurivorosphaerella nawae*

**Oliul Hassan[1], Taehyun Chang[1]\*, Abul Hossain[2]**

**1** Department of Ecology & Environmental System, College of Ecology & Environmental Sciences, Kyungpook National University, Sangju-si, Gyeongsangbuk-do, Korea (Republic of), **2** Departmet of Biochemistry, Memorial University of Newfoundland, St. John's, NL, Canada

\* thchang@knu.ac.kr

**Data Availability Statement:** All relevant data are within the manuscript and Supporting Information files.

## Abstract

Circular leaf spot, caused by the ascomycetous fungus *Plurivorosphaerella nawae* (= *Mycosphaerella nawae*), is the most problematic fungal disease of persimmon worldwide. In Korea, persimmon exposed to *P. nawae* inoculum (ascospores) from May to August shows visible circular leaf spot disease symptoms from the end of August to early September. It is important to identify factors affecting this long latent period. The objective of this study was to elucidate the relation between the development of symptom of circular leaf spot and the content of phenolics compounds and vitamin C as well as the antioxidant activities in leaves. Healthy leaves (both young and old) and infected leaves of circular leaf spot-susceptible persimmon cultivar were harvested in 2016. The content of phenolics (total phenols, flavonoids, and tannins) and vitamin C, and their antioxidant activities were analyzed in all types of leaves. Compared with the asymptomatic leaves (old) and the asymptomatic parts of the infected leaves, the symptomatic parts of the infected leaves, symptomatic leaves, and asymptomatic young leaves showed significantly higher content of phenolics and vitamin C, and higher antioxidant activities. Disease incidence and severity were estimated for older leaves (emerged in early May) and younger leaves (emerged at the end of June) in 2017 and 2018. The AUDPC was higher in old leaves than younger leaves. The disease progression was much faster and severe in the older than in the younger leaves. Similar results were found in field experiments. Higher content of phenolics and antioxidant activities in the younger leaves may contribute to circular leaf spot resistance in persimmon. Furthermore, accumulation of phenolics and antioxidant activity in the infected leaves is a post-infection response and the first stage of the defense mechanism.

## Introduction

Circular leaf spot of persimmon (CLSP), caused by *Plurivorosphaerella nawae* (basionym:- *Mycosphaerella nawae*), is a very problematic disease, causing leaf necrosis, premature leaf fall,

**Funding:** This study was supported by a grant from the Cooperative Research Program for Correspondence Competitiveness Improvement Technology Development (Project No. PJ01169703) of the Rural Development Administration, Korea. No authors received a salary from the funder.

**Competing interests:** The authors have declared that no competing interests exist.

early ripening of fruits, and fruit abscission[1,2,3,4]. It is a foliar disease affecting persimmon worldwide, including South Korea, Japan, and Spain [2,3,4]. The disease cycle of CLSP has been described by several studies in Japan, Korea, and Spain [2,5,6]. The CLSP fungus survives during the winter as small black fruiting bodies (pseudothecia) in leaf litter. In Korea, this black fruiting body develops in the back side of the spots on persimmon leaves from October to November in different shapes, such as spherical ovoid, oval, and flask- and pear-type [6,7]. Pseudothecia become sclerotium-like before extreme winter and increases in the following years in the middle or late April [7,8]. The ascospores (sexual spores) in pseudothecia reach maturity by the end of April as temperature rises in the spring [7,8]. The ascospores get forcibly ejected from pseudothecia into the air and germinate on leaf surface. After a long quiescent period, typical CLSP symptoms appear on leaves at the end of August [1,7]. Owing to the long quiescent period, it is difficult to control CLSP at the initial stages of infection. In severe case CLSP is responsible 100% yield loss [1]. Persimmon growers are managing this disease through treatment with multiple fungicides (benomyl, mancozeb and trifloxystrobin). Resistant or tolerant cultivars have not been developed in Korea yet.

Phenolic compounds are secondary metabolites and are ubiquitous in all types of plants [9]. The functions of phenolic compounds are pigmentation of flowers and fruits, plant growth, incorporation of attractive substances to accelerate pollination, and seed dispersal [10,11]. In addition, phenolic compounds play an important role in plants defense against pathogens [12,13]. The phenolic compounds act as pesticides against plant-invading pathogens including bacteria, fungi, and nematodes [12]. The oxidation products of phenolic compounds are responsible for cytotoxicity, which is the basis for plant disease resistance [14,15]. Involvement of some phenolic compounds like flavan-3-ols in injury and stress resistance is proposed [15]. Anti-pathogenic activities of phenolic acids, flavonoids, tannins, etc., are well known [16,17]. The leaves of persimmon are used in preparing green tea because of the presence of different nutritious elements including flavonoids, tannins, and ascorbic acid [18,19,20]. Numerous scientific reports suggest the involvement of flavonoids in UV-scavenging, fertility, and plant defense against pathogens [15,21,22]. There is a positive correlation between the levels of flavonoids in the leaves and resistance against fungal pathogens, especially *Venturia inaequalis* [23]. Higher quantities of phenolic compounds have been reported in *V. inaequalis*-resistant apple cultivars than in the susceptible apple cultivars [24]. Phenolic compounds have also been reported as the most influential secondary metabolites in determining the resistance of pearl millet plants to *Sclerospora graminicola*, cowpea to *Black eye cowpea mosaic virus*, and raspberry to *Didymella applanata* and *Paraconiothyrium fuckelii* [17,25,26]. Tannins, another class of secondary metabolites, also influence plant resistance to pathogens and herbivores [27,28,29]. Tannins can inhibit spore germination and germ tube growth of *Verticillium albo-atrum* [29]. Resistance of common beans against *Colletotrichum lindemuthianum*, *Pseudocercospora griseola*, and *Xanthomonas campestris* is associated with higher tannin levels in the seed coat [27]. Ascorbic acid (vitamin C) is a major antioxidant compound associated with plant antioxidant-defense systems [30]. Vitamin C is involved directly or indirectly in protecting plants against pathogen invasion by interacting with the key components of a complex network regulating both basal and induced resistance in different pathosystems [31]. In coordination with glutathione (GSH) and important enzymatic antioxidants, vitamin C create redox environment in plant, which regulating diverse defense pathways such as the expression of defense genes, the strengthening of cell walls, and the modulation of defense-hormonal signaling networks [31].

There are some reports on the antioxidant properties and efficacies of persimmon leaves [20,32,33]. However, the effect of the antioxidant properties on CLSP development has been inadequately studied. Therefore, the objective of this study was to investigate the patterns of

occurrence of some phenolics and vitamin C in the healthy and CLSP infected leaves to elucidate the relation between the levels of these components and the development of CLSP.

## Materials and methods

### Plant material and field conditions

The study was carried out in 2016 using the leaves of five-year-old persimmon trees (cultivar 'Dongsi' were grafted onto rootstock, *Diospyros lotus*), growing in Sangju, South Korea. The orchard was irrigated by using under-canopy sprinkler (Low Elevation Spray Application) irrigation system. The experimental orchard is known CLSP hot-spots and fungicides were not sprayed before and during the study period. Fully developed symptomatic and asymptomatic leaves were collected from the persimmon tree twice, on May 27 (flowering stage) and September 20 (after circular leaf spot disease symptom expression). Phenolic compounds (total phenols, total flavonoids, and total tannins), vitamin C, and antioxidant activities (elucidated by estimating the quenching of 2,2-diphenyl-1-2picrylhydrazyl (DPPH) and 2,2'-azino-bis (3-ethylbenzothiazoline-6-sulphonic acid) (ABTS) activities) were analyzed in the asymptomatic leaves collected on May 27 (H1) and September 20 (H2), symptomatic leaves (collected on September 20) with 75–100% disease severity (D), and asymptomatic (1/2 H) and symptomatic parts (1/2 D) of the infected leaves (Fig 1).

### Processing and preparation of aqueous extracts of the collected persimmon leaves

Aqueous extract of the persimmon leaves was processed and prepared according to the procedure described by Hossain et a. [34]. The collected leaves were air dried at 100˚C for 30 min after blanching at 100˚C for 2 min. The dried leaves were homogenized to fine powder ($\leq$ 1 mm particle sized) using a Waring commercial blender (PBB25 Stainless Steel Blender, Stamford; Connecticut, USA). Leaf powder was soaked in distilled water in the ratio of 1:10 (w/w) at 90˚C for 60 min to extract the leaf contents. The extracts were filtered through Whatman no. 4 filter paper (110 mm $\emptyset$, Whatman International Limited, Buckinghamshire, U.K.). The filtrates were collected and stored at 4˚C until further use.

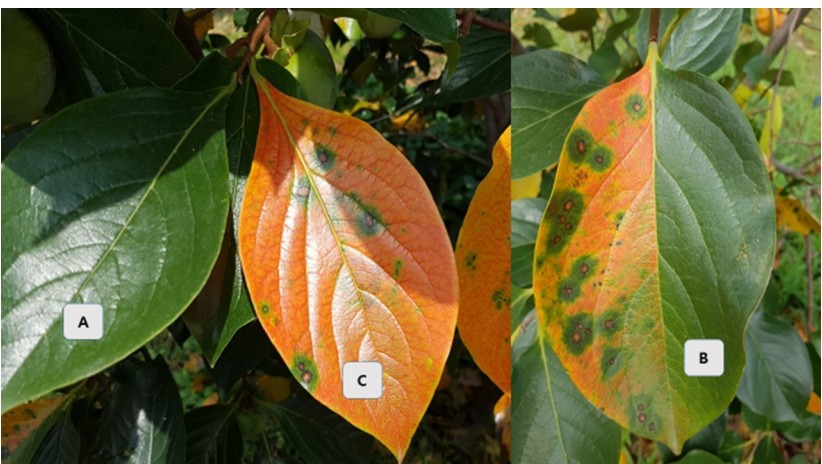

**Fig 1. Different types of leaves.** A. asymptomatic leaf at two collection time point (H1, H2). B. symptomatic leaf (D). C. Diseased containing 1/2 healthy part (1/2 H, 1/2 D).

## Determination of total phenolic content

Total phenolic content was estimated with Folin-Ciocalteu's reagent according to the procedure described by Singleton and Rossi [35], with slight modifications. Briefly, 0.2 mL of the filtrate sample solution was reacted with 1 mL of 10% Folin-Ciocalteu's reagent (Sigma Chemicals, St Louis, MO, USA) in a flask. After 10 min, 0.8 mL of 7.5% sodium carbonate ($Na_2CO_3$) was added to the reaction mixture and was incubated at room temperature (25°C) for 2 h. The absorbance was measured at 765 nm using a UV-VIS spectrophotometer (Shimadzu Corp., Kyoto, Japan). Serially diluted gallic acid solution (0–200 mg/L) was used as a standard. Results were expressed as milligram of gallic acid equivalents per gram of dry matter (mg GAE/g).

## Determination of total flavonoid content

Total flavonoid content of the collected persimmon leaves was determined by modified colorimetric methods using catechin as a standard compound [36]. Catechin is well known flavonoid used as standard [20]. Catechin solution (0, 12.5, 25, 50, 100 mg/L) was used as standard. A volume of 0.25 mL of the filtrate was taken in a test tube along with 1 mL of distilled water. After adding 75 μL of a 5% sodium nitrite ($NaNO_2$) solution, the mixture was allowed to stand for 6 min. Subsequently, 150 μL of 10% aluminum chloride ($AlCl_3$) was added and incubated for 5 min, after which 0.5 mL of 1 M sodium hydroxide (NaOH) was added, followed by the addition of 2 mL distilled water. Finally, the absorbance was read against a blank at 510 nm using a spectrophotometer (Shimadzu Corp., Kyoto, Japan). Total flavonoid content was expressed as milligram of catechin equivalent per gram of dry matter (mg CE/g).

## Determination of tannin content

Total tannin content was estimated according to the modified method of Broadhurst and Jones [37], as described by Xu and Chang [38], using catechin as a reference compound. A mixture of 0.5 mL of the persimmon leaf filtrate, 3 mL of a 4% vanillin in methanol, and 1.5 mL of hydrochloric acid (conc.) was prepared and incubated at 25°C for 15 min. Subsequently, the absorbance was read at 500 nm using a spectrophotometer (Shimadzu Corp.) against a methanol blank. Results were expressed as milligram of catechin equivalent per gram of dry matter (mg CE/g).

## DPPH radical scavenging activity

Quenching of 2,2-diphenyl-1-picrylhydrazyl (DPPH) radical scavenging activity by persimmon leaf extract was evaluated according to the method described by Shahidi et. al. [39] with some modifications. Persimmon leaf extract was diluted (1 mL extract/5 mL methanol), and 3.9 mL of DPPH solution in methanol (methanol: water = 80:20; final content = 0.025 mg/mL) was added to 0.1 mL of diluted persimmon leaf extract. The mixture was vortexed for 1 min and allowed to stand at the ambient temperature ($\geq$ 25°C) for 30 min. Subsequently, the absorbance was measured at 510 nm using a spectrophotometer (Shimadzu Corp.). The percentage of antioxidant activity was calculated according to the following equation:

$$\% \text{ inhibition of DPPH} = [(A_{control} - A_{sample})/A_{control}] \times 100. \tag{i}$$

Where, $A_{control}$ is the absorbance of the DPPH solution only.

## ABTS radical-scavenging activity

The ABTS scavenging activity of persimmon leaf extract was determined by methods of Thaipong et al. [40] with some modifications. The working solution was prepared by mixing equal quantities of 7.4 mM ABTS solution and 2.6 mM potassium persulfate solution. After incubating it in the dark at room temperature (25°C) for 12 h, the working solution was diluted by adding 1 mL of ABTS mixture to 24 mL of ethanol, to obtain an absorbance of 1.5 ± 0.02 at 734 nm using a spectrophotometer. Fresh ABTS solution was used for each assay. Finally, 0.3 mL of leaf extract was added with 2.7 mL of the ABTS working solution. Then the absorbance was measured after 7 min at 734 nm. The results were expressed as percentage (%) and calculated according to the formula:

$$\text{Scavenging activity (\%)} = 100 - [(\text{O.D. of sample}/\text{O.D. of control}) \times 100]. \quad \text{(ii)}$$

Where, O.D. is the observed data and control represent the absorbance of the ABTS solution.

## Determination of total vitamin C content

Vitamin C was quantified according to the modified simple UV-spectrophotometric method described by Khan et al. [41]. Briefly, collected leaves were freeze dried for 72 h in a vacuum freeze dryer at -50°C and blended for making powder [20,34]. 5 g of persimmon leaf powder was homogenized with 25 mL of 3% metaphosphoric acid-8% acetic acid (MPA-AA) solution, transferred into a 50 mL volumetric jar, and shaken gently to homogenize the solution. It was diluted up to the mark using the same MPA-AA solution. The dilutes solution was then filtered using Buchner funnel with paper filter (Buchner funnel vacuum filtration). The filtered clear sample solution was collected and 4–5 drops of bromine water was added until the solution developed red color. For eliminating the extra bromine and obtaining a clear solution, two to four drops of 10% thiourea ($CH_4N_2S$) solution was added. Serially diluted standard ascorbic acid solutions (0–250 ppm) were prepared from 500-ppm stock solution (Signa-Aldrich). Then 1 mL of 2,4-dinitrophenyl hydrazine (2,4-DNPH) solution was added to all the standard solutions and the oxidized sample solution. The standard, sample, and blank solutions were incubated in a water bath at 37°C for 3 h. After incubation and cooled on ice bath, all the solutions (standard, sample, and blank) were treated with 1 ml 85% sulfuric acid. Then the absorbance was measured using a spectrophotometer at 521 nm.

## Disease evaluation in field experiment

Disease severity was evaluated under field and control conditions (described below). For the experiment under field conditions, five trees (one-tree plots) of cultivar Dongsi grafted on rootstock *Diospyros lotus* were randomly selected in the experimental area in Kyungpook National university farm (KNU farm) and Woeanum orchards (WO), Sangju, Korea. The age of persimmon in KNU farm and WO were around 4 and 5 years, respectively. The orchard was irrigated by using under-canopy sprinkler (Low Elevation Spray Application) irrigation system. No fungicide was used in either orchard. The disease severity was estimated in August 23 and 30; September 6, 13, and 20; and October 4 and 11 in both years, 2017 and 2018. Ten shoots sprouted in early May (old shoots) and ten shoots sprouted around mid- June (new shoots) were randomly selected for the disease assessment in each tree. All the leaves on old shoots (old leaves) (approx. 50 leaves per tree) and new shoots (new leaves) (approx. 50 leaves per tree) were rated on the following scale: 0 = no disease; 1 = 1% of leaf area affected; 2 = 5%

of leaf area affected; 3 = 10% of leaf area affected 4 = 25% of leaf area affected; 5 = 50% of leaf area affected; and 6 = 75% of leaf area affected and defoliation occurred.

Disease severity (DS) was calculated using Townsend-Heuberger's formula [42]:

$$DS\ (\%) = \frac{\sum nv}{NV} \times 100. \tag{iii}$$

Where, n = degree of infection on the 6-grade scale, v = number of leaves per category, V = total number of leaves assessed, N = the highest degree of infection.

Area under the disease progress curve (AUDPC) was estimated from estimated percent of disease severity recorded at different times in R package 'stats' (version 3. 4.1) [43]. AUDPC was calculated using midpoint formula [44]:

$$AUDPC = \sum_{i=1}^{Ni-1} \left( \frac{yi + yi + 1}{2} \right) (t_{i+1} - t_i). \tag{iv}$$

Where t = the time of each reading, y = the percentage of disease severity at each reading and n = the number of readings and t = time.

Disease progression curve was constructed using AUDPC as a summary of disease severity to compare the differences between the old and younger leaves.

## Experiment under controlled condition

The effect of leaf age on CLSP development was evaluated on approx. 2-year old persimmon seedlings. For this assay, potted persimmon seedlings (cultivar 'Dongsi' were grafted onto rootstock, *Diospyros*) were managed in an isolated place (Kyungpook National University Sangju campus). Potted seedlings with two leaf ages were developed by pruning of old leaves. Seedling with first emerging leaves (between end of the April and early May) were treated as old leaves. Seedling with emerging leaves around 15th June were considered as younger leaves. The seedlings were inoculated by spraying with fresh ascospore suspension ($1 \times 10^6$ ml$^{-1}$) two times. Seedlings were inoculated on 23rd and 28th of June. The primary spore suspension was obtained by soaking overwintering diseased leaves (only lesion) in ultra-pure water. The concentration of the primary spore suspension was measured using a hemocytometer and then working spore suspension was prepared. Seven seedlings per treatment were used. Disease severity were assessed on September 8, 15, 22 and 29; and October 10 on the scale described above. AUDPC was calculated as described above.

## Statistical analysis

All data were analyzed using the Statistical Analysis System (SAS 9.3) program. Data were then analyzed by parametric tests such as one-way ANOVA and each treatment were compared using least significant difference test (LSD) ($\alpha = 0.05$). The data are presented as the mean ± standard deviation of triplicate and pentaplicate measurements. The total accumulated AUDPC until $t = t_k$ was also analyzed using ANOVA. Then, the total accumulated AUDPC each treatment (old leaves and younger leaves) were compared using least significant difference test (LSD) ($\alpha = 0.05$).

## Results

### Total phenolic compounds, flavonoid, tannin, and vitamin C contents

Total content of phenolics, flavonoids, tannins, and vitamin C in different types of persimmon leaves are shown in Figs 2 and 3. The total phenolic content in the H1 leaves ranged from

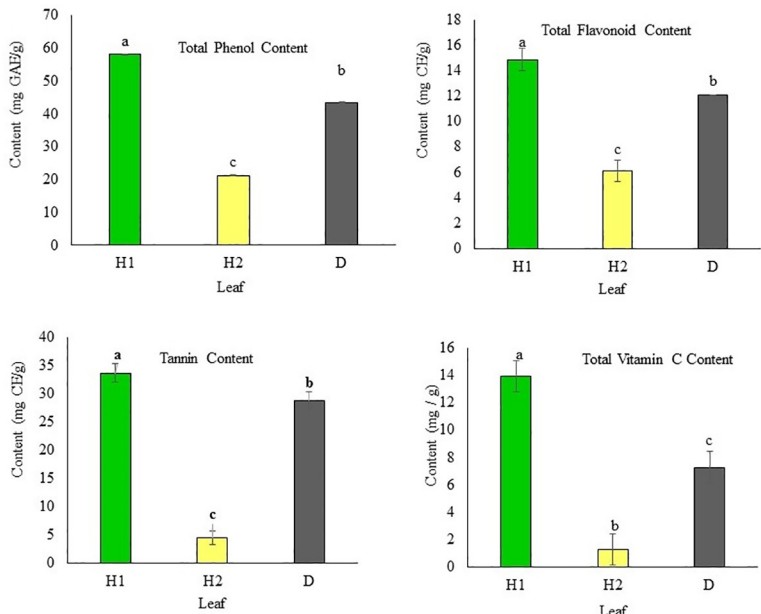

**Fig 2. Phenolic compounds and vitamin C content in persimmon leaves.** H1: asymptomatic leaf (mid-May); H2: asymptomatic leaf (beginning of September); and D: symptomatic leaves infected with *Plurivorosphaerella nawae*. Different letters indicate that the means are significantly different at 95% confidence level.

58.015 to 58.16 mg GAE/g (mean = 58.06 mg GAE/g); in the H2 leaves, from 21.20 to 21.34 mg GAE/g (mean = 21.24 mg GAE/g); in the D leaves, from 43.32 to 43.77 mg GAE/g (mean = 43.52 mg GAE/g); in the 1/2H leaves, from 19.53 to 19.68 mg GAE/g (mean = 19.58 mg GAE/g); and in the 1/2D leaves, from 37.25 to 37.40 mg GAE/g (mean = 37.30 mg GAE/g). There was a significant difference ($p < 0.01$) among the mean total phenolic content of the H1, H2, and D leaves and between those of the 1/2 H and 1/2 D leaves.

The total flavonoid content in the H1 leaves ranged from 14.16 to 15.83 mg CE/g (mean = 14.86 mg CE/g); in the H2 leaves, from 5.41 to 7.08 mg CE/g (mean = 6.11 mg CE/g); in the D leaves, from 12.09 to 12.07 mg CE/g (mean = 12.08 mg CE/g); in the 1/2 H leaves, from 5.83 to 7.08 mg CE/g (mean = 6.38 mg CE/g); and in the 1/2 D leaves, from 10 to 10.83 mg CE/g (mean = 10.41 mg CE/g). There was a significant difference ($p < 0.01$) among the mean total flavonoid content of the H1, H2, and D leaves and between those of the 1/2 H and 1/2 D leaves.

The tannin content in the different types of leaves showed a trend that was similar to that of the phenolic and flavonoid content. The tannin content in the H1 leaves ranged from 32.38 to 35.46 mg CE/g (mean = 33.66 mg CE/g); in the H2 leaves, from 3.15 to 5.46 mg CE/g (mean = 4.43 mg CE/g); in the D leaves, from 27 to 30.07 mg CE/g (mean = 28.79 mg CE/g); in the 1/2 H leaves, from 2.84 to 3.46 mg CE/g (mean = 3.13 mg CE/g); and in the 1/2 D leaves, from 18.53 to 19.30 mg CE/g (mean = 19.05 mg CE/g). There was a significant difference ($p < 0.01$) among the mean tannin content of the H1, H2, and D leaves and between those of the 1/2 H and 1/2 D leaves.

Vitamin C content in the H1 leaves ranged from 12.6 to 14.6 mg CE/g (mean = 13.93 mg CE/g); in the H2 leaves, from 0.6 to 2.6 mg CE/g (mean = 1.27 mg CE/g); in the D leaves, from 6.56 to 8.7 mg CE/g (mean = 7.27 mg CE/g); in the 1/2H leaves, from 0.6 to 2.46 mg CE/g (mean = 1.93 mg CE/g); and in the 1/2D leaves, from 4.53 to 6.30 mg CE/g (mean = 5.27 mg

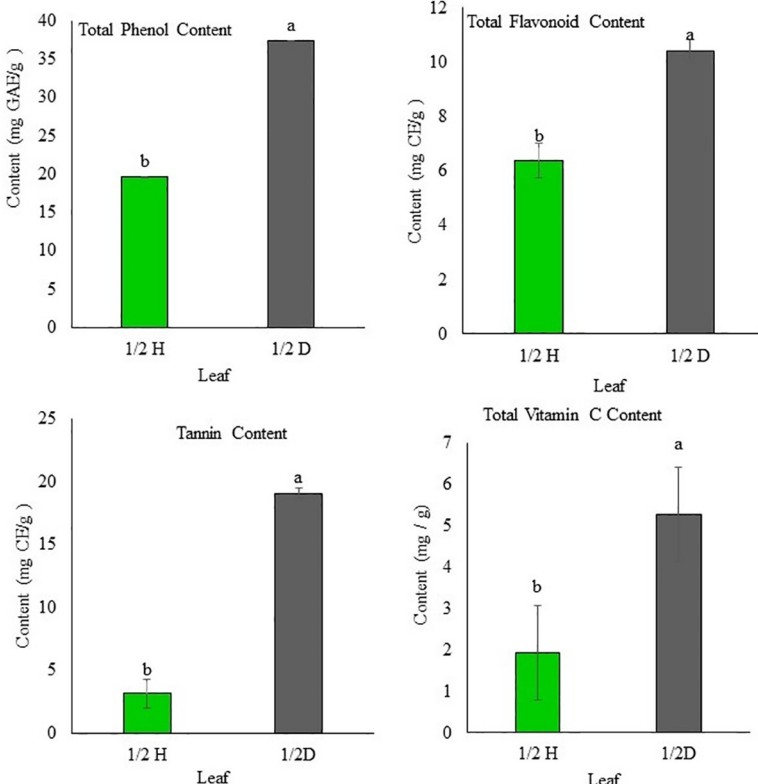

**Fig 3. Phenolic compound and vitamin C content in 1/2 H: asymptomatic part of infected leaf and 1/2 D: symptomatic part of infected leaf.** Different letters indicate that the means are significantly different at 95% confidence level.

CE/g). There was a significant difference ($p < 0.01$) among the mean vitamin C content of the H1, H2, and D leaves and between those of the 1/2 H and 1/2 D leaves.

## Antioxidant activities

The DPPH and ABTS radical-scavenging activities of the different types of persimmon leaves are shown in Fig 4. The DPPH radical-scavenging activity in the H1 leaves ranged from 42.11 to 42.65% (mean = 42.28%); in the H2 leaves, from 26.50 to 27.22% (mean = 26.8%); in the D leaves, from 34.30 to 34.48% (mean = 34.41%); in the 1/2 H leaves, from 26.50 to 27.04% (mean = 26.67%) and in the 1/2 D leaves, from 34.30 to 34.48% (mean = 34.42%).

The ABTS radical-scavenging activity in the H1 leaves ranged from 75.24.11 to 75.37% (mean = 75.31%); in the H2 leaves, from 34.76 to 35.32% (mean = 35.02%); in the D leaves, from 65.38 to 65.84% (mean = 65.69%); in the 1/2 H leaves, from 34.41 to 34.67% (mean = 34.52%); and in the 1/2 D leaves, from 56.31 to 56.57% (mean = 56.47%). There was a significant difference ($p < 0.01$) among the mean DPPH and ABTS radical-scavenging activities of the H1, H2, and D leaves and between those of the 1/2 H and 1/2 D leaves.

## Disease progress

The AUPDC in the old leaves was significantly higher than that of the new leaves in both the orchards in 2017 and 2018 (Fig 5). Both year early disease initiation, faster rates of disease development, and higher final disease severities were recorded for old leaves (Fig 5). First signs

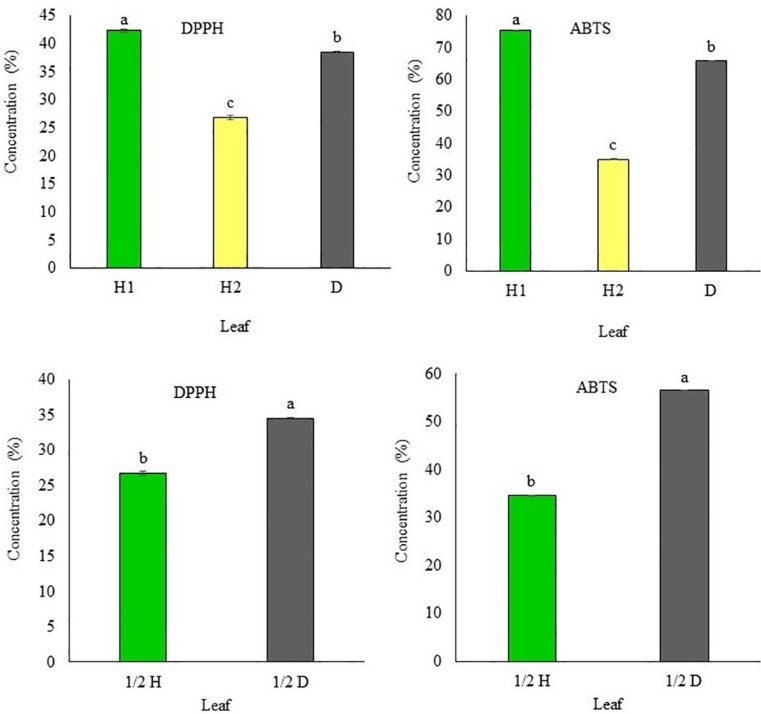

**Fig 4. Antioxidant activity in H1: asymptomatic leaf (early stage); H2: asymptomatic leaf (collected during the infection period); and D: symptomatic leaf; 1/2 H: asymptomatic part of infected leaf; 1/2 D: symptomatic part of the leaf infected with *Plurivorosphaerella nawae*.** Different letters indicate that the means are significantly different at 95% confidence level.

of CLSP appeared in the old leaves at the end of August, and high disease severity occurred in mid-October. In both years, the disease progress curves for old leaves reached a level ranging from 53 to 98% severity, while for the younger leaves reached severity levels ranging from 3% to 28% (Fig 5).

Under controlled conditions, signs of CLSP appeared in the old leaves at the early September while CLSP appeared in the new leaves in mid-September. As similar with the result of field experiment, AUPDC in old leaves was significantly higher than younger leaves (Fig 6). The disease progress curves showed that diseases severity recorded for old leaves reached at 99%; for the younger leaves, it reached about 32% (Fig 6).

## Discussion

A number of fungal disease challenges persimmon cultivation. Among them, CLSP caused by *P. nawae* is a widespread and destructive disease. *Plurivorosphaerella nawae* infection does not kill persimmon trees, but decreases the yield by causing leaf necrosis, premature leaf fall, early ripening of fruits, and fruit abscission [3,4]. To limit disease severity, plants synthesize phenolic compounds (phenols, phenolic acids, flavonoids, tannins, etc.) [16,17]. The green leaves of persimmon are rich in flavonoids, tannins, ascorbic acid [18,19,20]. In this study, we investigated the roles of total phenol content, total flavonoids, tannins, and vitamin C in the defense of persimmon against *P. nawae* infection. Our results showed that the asymptomatic leaves (H1) collected at the end of May contained significantly higher content of total phenols, total flavonoids, tannins, and vitamin C, when compared with the asymptomatic leaves (H2) and symptomatic leaves (D) collected during the disease period. Furthermore, the asymptomatic

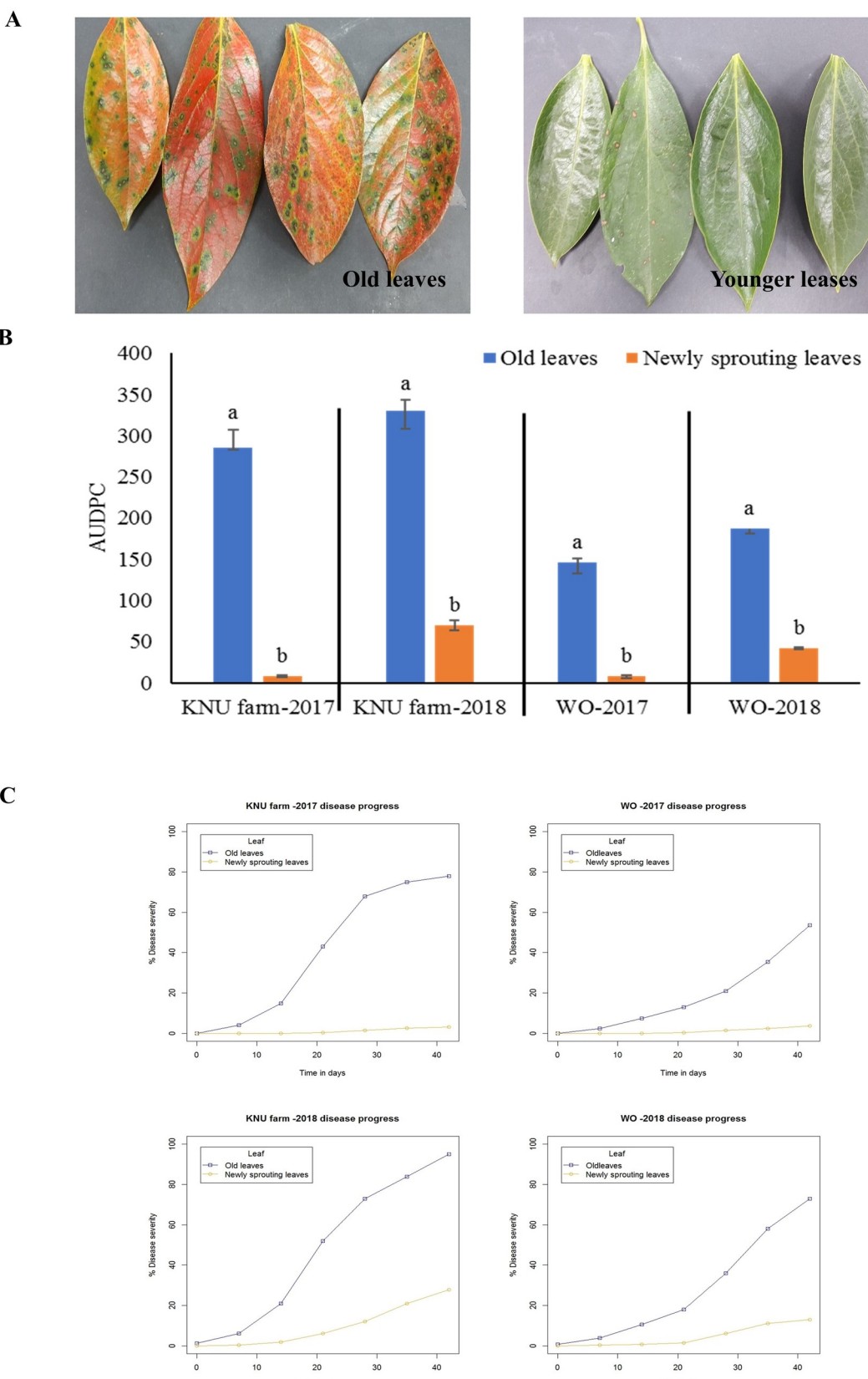

**Fig 5. Circular leaf spot in older and younger leaves of persimmon under field condition.** A. Symptoms of CLSP in older and younger leaves. B. The AUDPC of ceircular leaf spot in older and younger leaves C. Area under the disease progress curve to compare disease severity between older and younger leaves at different orchards in 2017 and 2018.

leaves (H2) contained significantly lower content of total phenols, total flavonoids, tannins, and vitamin C, when compared with the symptomatic leaves (D) collected during the disease period (Fig 2). Anti-oxidant activities (DPPH and ABTS radical-scavenging activities) were also found to be significantly high in H1, followed by D and H2 (Fig 4). The phenolic compounds, vitamin C, and both the anti-oxidant activities were significantly higher in the symptomatic part (1/2 D) than in the asymptomatic part (1/2 H) of the same leaf (Fig 3). The high content of total phenols, total flavonoids, tannins, and vitamin C in the H1 leaves are consistent with the findings of Jung and Jeong [33], who found that the extract of persimmon leaves harvested in May showed the highest levels of total phenolic compounds, flavonoids, and DPPH free radical-scavenging activities, compared with the levels in the extract of leaves collected in other months. As the content of phenolic compounds in persimmon leaves decrease after May, this study found the lowest level of phenolic compounds in the H2 leaves. Interestingly, higher levels of phenolic compounds and vitamin C were observed in the symptomatic leaves (D) than in the H2 leaves, although both types of leaves were collected at the same time. This can be explained by basal resistance, which is triggered by the attack of pathogens. After recognizing the microbe-associated molecular patterns (MAMPs), plant cells become fortified against pathogen attacks by accumulating chemicals like terpenoids, phenolics, and alkaloids [45]. The accumulation of different phenolic compounds in fungus-infected leaves and fruits has been shown in previous studies [23, 46,47]. The findings of this study, with regard to higher accumulation of phenolic compounds and vitamin C in the symptomatic part (1/2 D) than in the asymptomatic (1/2 H) part of the same leaf, can be explained by the higher accumulation of phenolic compounds in the asymptomatic plant cells surrounding the wounded or infected cells. The defense compound produced in the damaged cells diffuse into the adjacent healthy cells. Hence, activities of many phenol-oxidizing enzymes in the adjacent healthy cells are stimulated [48,49]. It has also been reported that synthesis and accumulation of phenolic compounds occurs between the circular leaf spot-affected tissue and the healthy tissue [23,49]. Higher levels of phenolic compounds in the infected parts can play a role in the protection of plant health [49]. There was a positive correlation between phenolic compound accumulation and antioxidant activity (not shown here), which is consistent with the findings of Hossain et al. [34].

Another important finding of this study is higher AUDPC for older leaves than younger leaves (Fig 5). Higher AUDPC means the more disease severity in old leaves. The disease progression was much faster and severe in leaves emerged in May compared with those emerged in June in both years. (Fig 5). Both disease intensity and severity were higher in the older leaves than in the younger leaves. The findings under field conditions were complemented by those under experimental conditions (Fig 6). According to the findings of this study, the effect of phenolic compounds and vitamin C on CLSP development is substantial. Higher content of pre-formed phenolic compounds in the younger leaves (H1) may contribute to the basal resistance of persimmon against circular leaf spot. High phenolic compound levels in plants may inhibit pathogen attachment, invasion, and infection [48, 50]. There are contradicting reports regarding the influence of environmental conditions on the infection process of *P. nawae*. On one hand, although the effect of specific temperature and moisture level have not been determined, a film of water and appropriate temperature have been reported to be required for infection development [1]. The question regarding the effect of different inoculum loads on

**A**

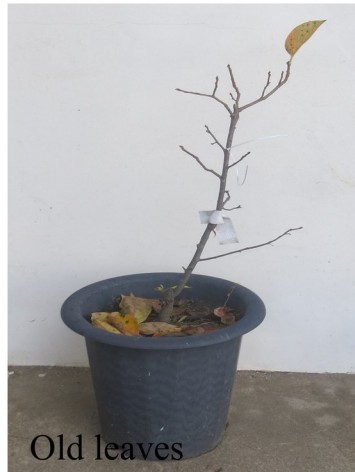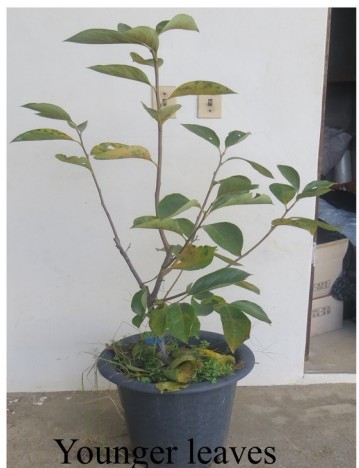

Old leaves

Younger leaves

**B**

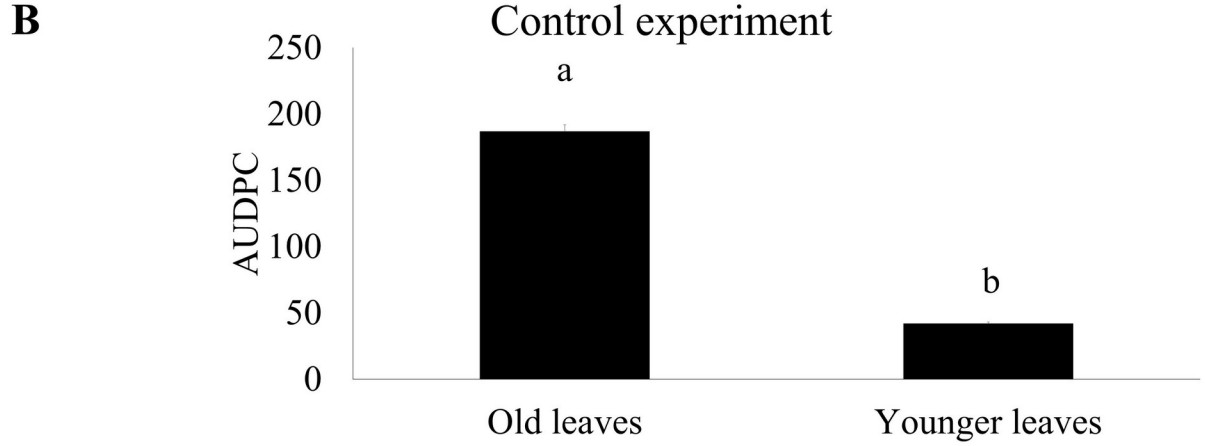

**C**

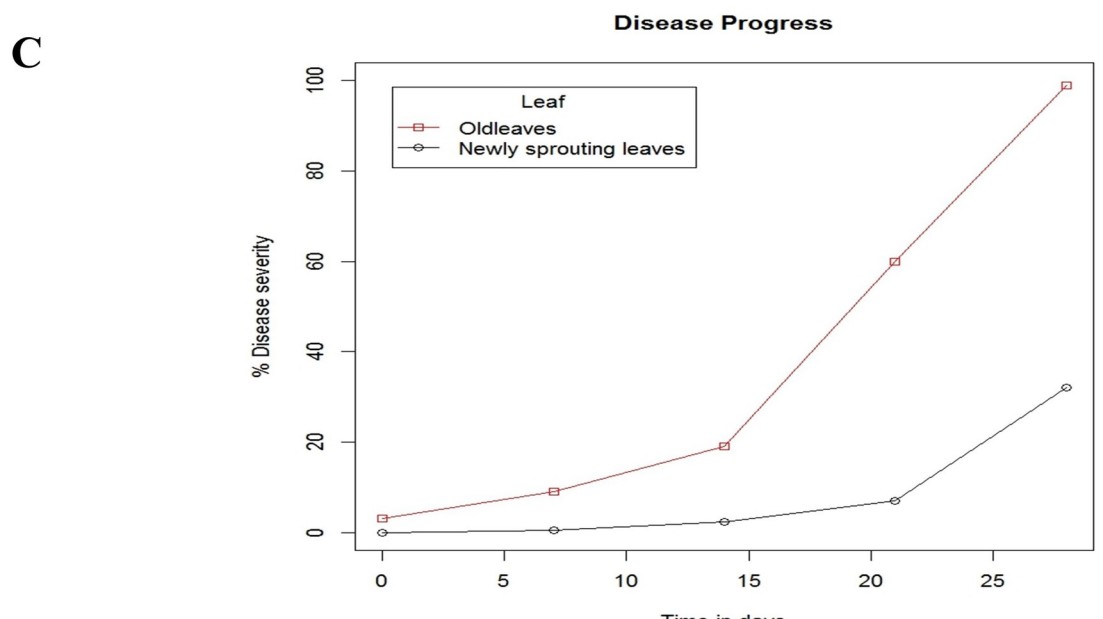

**Fig 6. Circular leaf spot in older and younger leaves of persimmon under control condition.** A. CLSP severity on older and younger leaves containing seedling. B. The AUDPC of ceircular leaf spot in older and younger leaves inoculated with ascospore of *P. nawae*. C. Area under the disease progress curve to compare disease severity between older and younger leaves.

leaves emerged in May in June may be explained by the findings of studies related to disease outbreak and inoculum dynamic of CLSP. In Korea, maximum number of ascospores detected from mid-June to mid-July [7]. No significant difference between infection rate of plants in May and plants in June [1]. In Spain, relatively low inoculum load in June were able to induce severe symptoms on trap plants [2]. Hence, the effect of different inoculum load may not be the critical factor for inducing severe symptoms.

Based on the results of this study, it can be hypothesized that phenolic compounds and vitamin C may be associated with the long quiescent period observed for *P. nawae*. In Korea, persimmon is generally exposed to the inoculum (ascospores) of *P. nawae* from May, but symptoms appear at the end of August or in early September [7,51]. A tentative relationship between leaf phenolic compound (flavonoids) content and *P. nawae* infection is shown in Fig 7.

From May to mid-July, flavonoid content in the leaves is very high and maximum airborne ascospores are detected. During this period, higher levels of pre-formed flavonoids in leaves may resist *P. nawae* infection. If the spores are able to infect the leaves, they may survive as endophytes. The distinct association of *P. nawae* with plants in forms other than as pathogens, such as endophytes and opportunistic pathogens, have been reported by some studies [4,52]. When persimmon leaves contain lower flavonoid content from the end of July to early August, *P. nawae* infect leaves and cause disease. Pre-existing association of *P. nawae* as an endophyte or as an opportunistic pathogen may be converted to a pathogenic association to cause the disease. After a short period, disease appears during August or early September. During the

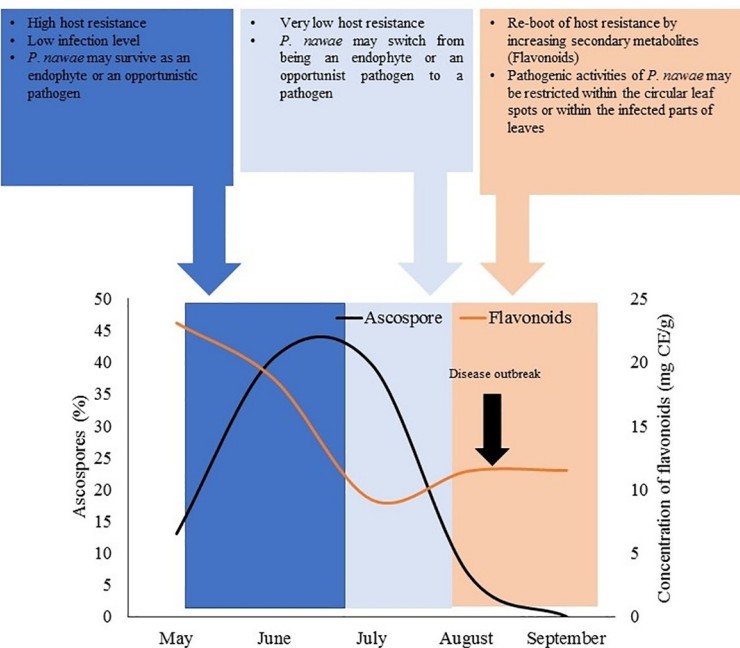

**Fig 7. Hypothetical relationship between *P. nawae* infection and flavonoid content of persimmon.** The trend line for ascospore dynamics is adopted from Hassan et al. [7]. The trend line for flavonoid content in persimmon leaf extracts is acquired from Jung and Jeong [33].

infection period, flavonoid content in persimmon leaves increases and induces the post-inflectional resistance to protect the healthy tissues of the plant. The relationship between phenolic compounds (flavonoids) and *P. nawae* infection was not validated with concrete evidence. A further study is needed for validating the proposed tentative relationship between phenolic compound content in leaves and *P. nawae* infection.

## Conclusion

One of the key and fascinating features of circular leaf spot is its long incubation period. Very little is known about the factors driving symptom expression in this disease. This study was on the biochemical factors influencing ontogenic susceptibility of persimmon leaves to circular leaf spot caused by *Plurivorosphaerella nawae* (*Mycosphaerella nawae*). Phenolic compounds and antioxidants were quantified in persimmon leaves in May and later in September in an affected orchard. Higher levels of these biochemical compounds were found in asymptomatic leaves in May (young leaves) than in asymptomatic leaves in September (old leaves). Symptomatic leaves in September had higher levels than asymptomatic leaves in September. Field observations in infected orchards showed that the disease progression was much faster and severe in leaves emerged in May compared with those emerged in June. It was confirmed by the experiment under control condition. The constitutive and induce phenolic compounds and vitamin C may explain the broad and unspecified prevention of plant diseases such as CLSP.

## Supporting information

**S1 Data. Secondary compounds obvs.**
(XLSX)

## Acknowledgments

We are very indebted to our lab mates who help during the period of study.

## Author Contributions

**Conceptualization:** Oliul Hassan.

**Data curation:** Oliul Hassan.

**Funding acquisition:** Taehyun Chang.

**Investigation:** Oliul Hassan, Abul Hossain.

**Methodology:** Oliul Hassan, Abul Hossain.

**Supervision:** Taehyun Chang.

**Writing – original draft:** Oliul Hassan.

**Writing – review & editing:** Taehyun Chang.

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
