## [Decision Letter · Decision Letter 0]

10 Feb 2020

PONE-D-20-01059

Changes in the secondary compounds of persimmon leaves as a defense against circular leaf spot caused by *Plurivorosphaerella nawae*

PLOS ONE

Dear Dr. Chang,

Thank you for submitting your manuscript to PLOS ONE. After careful consideration, we feel that it has merit but does not fully meet PLOS ONE’s publication criteria as it currently stands. Therefore, we invite you to submit a revised version of the manuscript that addresses the points raised during the review process.

The main drawback of the experimental design are the analytical methods. Authors are addressed to carefully consider concerns of Reviewer #1.

We would appreciate receiving your revised manuscript by Mar 26 2020 11:59PM. To enhance the reproducibility of your results, we recommend that if applicable you deposit your laboratory protocols in protocols.io, where a protocol can be assigned its own identifier (DOI) such that it can be cited independently in the future. For instructions see: http://journals.plos.org/plosone/s/submission-guidelines#loc-laboratory-protocols

We look forward to receiving your revised manuscript.

Kind regards,

Branislav T. Šiler, Ph.D.

Academic Editor

PLOS ONE

Journal Requirements:

"This study was supported by a grant from the Cooperative Research Program for Correspondence Competitiveness Improvement Technology Development (Project No. PJ01169703) of the Rural Development Administration, Korea."

"The funders had no role in study design, data collection and analysis, decision to publish, or preparation of the manuscript"

Reviewers' comments:

Reviewer's Responses to Questions

**Comments to the Author**

1. Is the manuscript technically sound, and do the data support the conclusions?

Reviewer #1: Partly

Reviewer #2: Yes

2. Has the statistical analysis been performed appropriately and rigorously? 

Reviewer #1: I Don't Know

Reviewer #2: Yes

3. Have the authors made all data underlying the findings in their manuscript fully available?

Reviewer #1: Yes

Reviewer #2: No

4. Is the manuscript presented in an intelligible fashion and written in standard English?

Reviewer #1: Yes

Reviewer #2: Yes

5. Review Comments to the Author

Reviewer #1: In the article, there is described a holistic response of persimmon leaves on the infection of P. nawae. The experiment is well designed but I have some doubts about analyzes of metabolites. In line 111 cited is a method that is not listed in the references. The drying and blanching at 100 °C could influence the content of phenolics and especially ascorbic acid. Furthermore, vitamin C includes L-ascorbic acid and dehydroascorbic acid, not only ascorbic acid. And a temperature range of 30 to 60 degrees C could result in the conversion of L-ascorbic acid to dehydroascorbic (DHAA), a very important reaction in regard to vitamin C degradation because DHAA could be easily converted to other compounds that do not have the biological activity of vitamin C (Munyaka et al., 2010, J Food Sci.; Herbig and Renard, 2017, Food Chem.).

When the units are mg/g we do not use the term “concentration” but “content”. The term “concentration” is used when the amount is present in volume.

Vitamin C - unifrom the writting

Line 105: define more specific when infected leaves were collected

Line 129: referencing is not appropriate

Line 130: a comma is missing

Line 157: why for Acontrol the DPPH solution was used and not blank (sample replaced by water+MeOH)?

Line 175: … filtered using Bucher. Explain more precisely what it is Bucher.

Line 206: what is meaning “r. AUDPC” ?

Line 318: What did you mean with “phenols” in the statement: “… phenolic compounds (phenols, phenolic acids, flavonoids, …)?

Line 332: referencing is not appropriate

References have to be checked detailed. Many of them are not correct used in the text. In the part References some are missing (Hossain et al., 2017), some are written twice (Mikulic-Petkovsek et al., 2011)…

Reviewer #2: In December 2018 I reviewed this same manuscript for the journal Plant Disease. Below you can find my two review reports. In this new version submitted to PlosOne the authors have included an additional experiment, which addresses my main concern at that time "To assure that the differences observed were in fact due to leaf age, potted plants with different leaf ages (pruning and/or cold storage may assist for this) should be inoculated at the same time and with the same inoculum concentration". I think now the paper can be accepted for publication.

-First review Plant Disease

In their manuscript, Hassan et al. presented a study on the biochemical factors influencing ontogenic susceptibility of persimmon leaves to circular leaf spot caused by Plurivorosphaerella nawae (Mycosphaerella nawae). Phenolic compounds and antioxidants were quantified in persimmon leaves in May and later in September in an affected orchard. Higher levels of these biochemical compounds were found in asymptomatic leaves in May than in asymptomatic leaves in September. Symptomatic leaves in September had higher levels than asymptomatic leaves in September. Field observations in infected orchards showed that the disease progression was much faster and severe in leaves emerged in May compared with those emerged in June. Laboratory inoculations suggested a negative relationship between leaf age and the duration of the incubation period. One of the key and fascinating features of circular leaf spot is its long incubation period. Very little is known about the factors driving symptom expression in this disease and thus this study is considered original and deserves consideration. However, serious limitations in the experimental design used were detected and the conclusions were considered highly speculative based on current evidence. Authors refer to healthy and diseased leaves, but studies were conducted in severely infested orchards, so is better they refer to asymptomatic and symptomatic leaves. The disease was more severe and progressed faster in leaves emerged in May compared with those emerged in June. The authors assume this is due to leaf age, with old leaves (emerged in May) being more susceptible. However, leaves emerged in May were indeed exposed to a much higher inoculum load and more infection events than those emerged in June, which escaped part of the infection period. Moreover, the conclusion than young leaves are less susceptible or even resistant to circular leaf spot contradicts current evidence. In Korea, trap plants with young leaves were exposed to field infections and developed severe symptoms of circular leaf spot (Kang et al. 1993 RDA J. Agric. Sci. 35:337-343; Kwon and Park 2004. Res. Plant Dis. 10:209-216). Studies with trap plants also indicated that, regardless of the period of infection, symptoms expression is somehow synchronized. Laboratory inoculations would certainly help to clarify this, but the experiment included in this manuscript is not convincing. Mycosphaerella nawae does not form ascospores in vitro and conidia are only sparsely produced. Authors have to describe and substantiate much better how they got the spore suspensions for inoculations. Also, how previous infections were not interfering with this experiment since leaves were collected from infested orchards. The use of living plants instead of detached leaves may be also better in this context. Methodologies such as GFP-transformed isolates would probably be necessary to dilucidate the incubation period of this interesting pathogen. In the view of this reviewer, the manuscript is not acceptable for publication in its present form. Nevertheless, resubmission is encouraged whether additional evidences will be compiled. More comments and suggestions are included in the attached pdf.

-Second review Plant Disease

The authors have clarified some issues and improved terminology through the text. Experimental details that were lacking in the previous version (e.g. orchard characteristics and inoculation method) are now included in the revised manuscript. Nevertheless, instead of providing additional evidences, the authors basically refuted the main criticisms highlighted by this reviewer. In the field experiments, the factors leaf age and exposure to inoculum were clearly confounded. To assure that the differences observed were in fact due to leaf age, potted plants with different leaf ages (pruning and/or cold storage may assist for this) should be inoculated at the same time and with the same inoculum concentration. In the laboratory experiments, I am still of the opinion that inoculations should be conducted using living plants instead of detached leaves, particularly when biochemical compounds will be quantified. In the revised manuscript, the authors indicated that detached leaves used for artificial inoculation were collected from an orchard previously treated with fungicides, which is a bit surprising. As noted in my previous review, I think the manuscript addresses a very interesting topic, but additional evidences are needed to support the conclusions drawn.

6. PLOS authors have the option to publish the peer review history of their article (what does this mean?). If published, this will include your full peer review and any attached files.

Reviewer #1: No

Reviewer #2: No

---

## [Author Response · Author response to Decision Letter 0]

12 Feb 2020

Dear reviewers 

Thank you very much for your query and suggestions.

We tried to address all the concerns of you.

The response of all point raised by reviewer 1 is given in attached file.

Thank you for your time.

Your best regard 

Chang

---

## [Editor Report · Decision Letter 1]

13 Feb 2020

PONE-D-20-01059R1

Changes in the secondary compounds of persimmon leaves as a defense against circular leaf spot caused by *Plurivorosphaerella nawae*

PLOS ONE

Dear Dr. Chang,

Thank you for submitting your manuscript to PLOS ONE. After careful consideration, we feel that it has merit but does not fully meet PLOS ONE’s publication criteria as it currently stands. Therefore, we invite you to submit a revised version of the manuscript that addresses the points raised during the review process.

In the previous review round, authors were addressed to meticulously check and revise the manuscript according to reviewers' comments. Reviewer #1 have raised major concerns about the methodology used, particularly related to the plant material handling and temperature-dependent degradation of L-ascorbic acid. More thorough discussion both in the main text and in "response to reviewers" were expected. The claims stated in the response to the reviewer's concerns have to be supported by a steady literature survey, whether the authors agree with the comments or provide a rebuttal.

There is no need to capitalize the term "vitamin".

Wrong figure legend provided for Figure 5 in the Figure legends list.

L143: no need to state the full compound name in the subtitle.

A brief conclusion part should be provided.

We would appreciate receiving your revised manuscript by Mar 29 2020 11:59PM. To enhance the reproducibility of your results, we recommend that if applicable you deposit your laboratory protocols in protocols.io, where a protocol can be assigned its own identifier (DOI) such that it can be cited independently in the future. For instructions see: http://journals.plos.org/plosone/s/submission-guidelines#loc-laboratory-protocols

We look forward to receiving your revised manuscript.

Kind regards,

Branislav T. Šiler, Ph.D.

Academic Editor

PLOS ONE

---

## [Author Response · Author response to Decision Letter 1]

13 Feb 2020

Dear Editor

Thank you very much for your query and suggestions.

We tried to address all the concerns of you.

Your best regard 

Chang

---

## [Decision Letter · Decision Letter 2]

26 Feb 2020

Changes in the secondary compounds of persimmon leaves as a defense against circular leaf spot caused by *Plurivorosphaerella nawae*

PONE-D-20-01059R2

Dear Dr. Chang,

We are pleased to inform you that your manuscript has been judged scientifically suitable for publication and will be formally accepted for publication once it complies with all outstanding technical requirements.

With kind regards,

Branislav T. Šiler, Ph.D.

Academic Editor

PLOS ONE

Additional Editor Comments (optional):

Reviewers' comments:

Reviewer's Responses to Questions

**Comments to the Author**

1. If the authors have adequately addressed your comments raised in a previous round of review and you feel that this manuscript is now acceptable for publication, you may indicate that here to bypass the “Comments to the Author” section, enter your conflict of interest statement in the “Confidential to Editor” section, and submit your "Accept" recommendation.

Reviewer #1: All comments have been addressed

2. Is the manuscript technically sound, and do the data support the conclusions?

Reviewer #1: Yes

3. Has the statistical analysis been performed appropriately and rigorously? 

Reviewer #1: Yes

4. Have the authors made all data underlying the findings in their manuscript fully available?

Reviewer #1: Yes

5. Is the manuscript presented in an intelligible fashion and written in standard English?

Reviewer #1: Yes

6. Review Comments to the Author

Reviewer #1: (No Response)

7. PLOS authors have the option to publish the peer review history of their article (what does this mean?). If published, this will include your full peer review and any attached files.

Reviewer #1: No

---

## [Editor Report · Acceptance letter]

28 Feb 2020

PONE-D-20-01059R2 

Changes in the secondary compounds of persimmon leaves as a defense against circular leaf spot caused by *Plurivorosphaerella nawae*

Dear Dr. Chang:

I am pleased to inform you that your manuscript has been deemed suitable for publication in PLOS ONE. Congratulations! Your manuscript is now with our production department. 

With kind regards,

on behalf of

Dr. Branislav T. Šiler 

Academic Editor

PLOS ONE